# NOTELA: A Generalizable Method for Source Free Domain Adaptation

## Abstract

Source-free domain adaptation (SFDA) is a compelling problem as it allows to leverage any off-the-shelf model without requiring access to its original training set and adapts it using only unlabelled data. While several SFDA approaches have recently been proposed, their evaluation focuses on a narrow set of distribution shifts for vision tasks, and their generalizability outside of that scope has not yet been investigated. We put those recent approaches to the test by evaluating them on a new set of challenging—due to extreme covariate and label shift—and naturally-occurring distribution shifts in the audio domain. We study the task of adapting a bird species classifier trained on focalized recordings of bird songs to datasets of passive recordings for various geographical locations. Interestingly, we find that some recent SFDA methods *underperform* doing no adaptation at all. Drawing inspiration from those findings and insights, we propose a new method that improves on noisy student approaches by adjusting the teacher's pseudo-labels through Laplacian regularization. Our approach enjoys increased stability and significantly better performance on several of our proposed distribution shifts. We then look back at SFDA benchmarks in the vision domain and find that our approach is competitive with the state-of-the-art there as well.

## 1 Introduction

Deep learning has made significant progress on a wide range of application areas. An important contributing factor has been the availability of increasingly larger datasets and models (Kaplan et al., 2020; Song et al., 2022). A downside of this trend is that training state-of-the-art models has also become increasingly expensive. This is not only wasteful from an environmental perspective, but also makes the training of such models inaccessible to some practitioners due to the prohibitive resources required, or potential difficulties with data access. On the other hand, directly reusing already-trained models is often not desirable, as their performance can degrade significantly in the presence of distribution shifts during deployment (Geirhos et al., 2020). Therefore, a fruitful avenue is designing *adaptation methods* for pre-trained models to succeed on a new target domain, without requiring access to the original (*source*) training data, i.e., "source-free". Preferably this adaptation can be performed *unsupervised*. This is the problem of *source-free domain adaptation* (SFDA) that we target in this work.

Several models have been proposed recently to tackle SFDA. However, we argue that *evaluation* in this area is a significant challenge in and of itself: We desire SFDA methods that are *general*, in that they can be used for different applications to adapt an application-appropriate pre-trained model to cope with a wide range of distribution shifts. Unfortunately, the standard evaluation protocol only considers a narrow set of shifts in computer vision tasks, leaving us with a limited view of the relative merits among different SFDA methods, as well as their generalizability. In this work, we address this limitation by studying a new set of distribution shifts. We expand on the existing evaluation methods, in order to gain as much new information as possible about SFDA methods. We also argue that we should target distribution shifts that are naturally-occurring. This maximizes the chances of the resulting research advances being directly translated into progress in solving real-world problems.

To that end, we propose to study a new set of distribution shifts in the *audio* domain. Specifically, we use a bird species classifier that was trained on a large dataset of bird song recordings as our pre-

Table 1: **Relationship of problem settings**. $x$ and $y$ denote inputs and labels, and $s$ and $t$ "source" and "target", respectively (note that in some cases, as in DG, $s$ might be a union of source domains / environments). For TTA and SFDA, the $*$ in their training data and loss reflects that they are entirely agnostic to how source training is performed, allowing the use of any off-the-shelf model.

|  | DA | DG | TTT | TTA | SFDA |
|---|---|---|---|---|---|
| Data used for training | $x^s, y^s, x^t$ | $x^s, y^s$ | $x^s, y^s$ | $*$ | $*$ |
| Data used for adaptation | $x^s, y^s, x^t$ | $x^s, y^s, x^t$ | $x^t$ | $x^t$ | $x^t$ |
| Training loss | $\mathcal{L}(x^s, y^s) + \mathcal{L}(x^s, x^t)$ | $\mathcal{L}(x^s, y^s)$ | $\mathcal{L}(x^s, y^s) + \mathcal{L}(x^t)$ | $*$ | $*$ |
| Adaptation loss | — | — | $\mathcal{L}(x^t)$ | $\mathcal{L}(x^t)$ | $\mathcal{L}(x^t)$ |

trained model. This dataset consists of *focalized recordings*, where the song of the identified bird is at the foreground of the recording. Our goal is to adapt this model to a set of passive recordings (*soundscapes*). The shift from focalized to soundscape recordings is substantial, as the recordings in the latter often feature much lower signal-to-noise ratio, several birds vocalizing at once, as well as significant distractors and environmental noise like rain or wind. In addition, the soundscapes we consider originate from different geographical locations, inducing extreme label shifts.

Our rationale for choosing to study these shifts is threefold. Firstly, they are challenging, as evidenced by the poor performance on soundscape datasets compared to focalized recordings observed by Goëau et al. (2018); Kahl et al. (2021). Secondly, they are naturally occurring and any progress in addressing them can support ecological monitoring and biodiversity conservation efforts and research. Finally, our resulting evaluation framework is "just different enough" from the established one: It differs in terms of i) the modality (vision vs. audio), ii) the problem setup (single-label vs multi-label classification), and iii) the degree and complexity of shifts (we study extreme covariate shifts that co-occur with extreme label-space shifts). Existing SFDA methods designed for vision tasks can also be evaluated using this new framework, since audio inputs are often represented as spectrograms and thus can be treated as images.

We perform a thorough empirical investigation of established SFDA methods on our new shifts. Interestingly, not only do some of the methods not improve the performance of the pre-trained model, they often *degrade* it. Studying this striking finding generates insights that lead us to make substantial modelling improvements. Notably, in the presence of extreme shifts, we observe that the confidence of the pre-trained model drops significantly and its calibration is poor as a result. This in turn poses a challenge for entropy minimization. While Noisy Student (Xie et al., 2020) copes best with this shift, it exhibits poor stability, necessitating careful early-stopping. This violates our assumption that labelled target data is unavailable. Our insight is that we can leverage the model's *feature space* as another "source of truth", as this space carries rich information about the relationship between examples. We propose Noisy Student Teacher with Laplacian Adjustment (NOTELA), a new model that enhances the Noisy Student model with a Laplacian regularizer. This model enjoys significantly increased performance on our new shifts as well as increased stability and resilience to model miscalibration. Closing the loop, we also evaluate our model on established vision benchmarks and observe that it is competitive with the state-of-the-art there; in fact largely surpassing it on two datasets.

## 2 RELATED WORK

Our discussion of related work is summarized in Table 1.

**Domain adaptation (DA).** DA assumes a setting in which labelled data is available for a source domain, and unlabelled data for a target domain. The goal is to maximize performance on the target domain. DA methods can be roughly divided into three types (Sagawa et al., 2022): *domain-invariant training* (also called *feature alignment*) aims to ensure that the features generated by the model for the source and target domain are indistinguishable by some metric (Sun et al., 2016; Sun & Saenko, 2016; Tzeng et al., 2014; Long et al., 2015; Ganin et al., 2016; Long et al., 2018;

Tzeng et al., 2017; Sankaranarayanan et al., 2018); *self-training* involves generating pseudo-labels for the unlabelled data (Xie et al., 2020); and *self-supervision* involves training an unsupervised/self-supervised model, later finetuned or jointly trained with a supervised objective (Shen et al., 2022).

**Source-Free Domain Adaptation (SFDA) and Test-time Adaptation (TTA).** These methods additionally assume that the source data itself is not available, e.g., because of resource, privacy, or intellectual property concerns. The distinction between SFDA and TTA is subtle: the latter is transductive, motivated by an online setup where adaptation happens on (unlabelled) target examples as they appear and evaluation is subsequently performed on the same examples. SFDA considers an offline adaptation phase and the adapted model is then evaluated on a different set of held-out examples. In practice, though, the methods developed for either are similar enough to be applicable to both. Related problems also include black-box (Zhang et al., 2021), online (Yang et al., 2020), continual (Wang et al., 2022b), and universal (Kundu et al., 2020) source-free domain adaptation.

Of the three types of DA methods discussed above, self-training most easily transfers to the SFDA and TTA settings (Liang et al., 2020; Kim et al., 2021), and in this work we focus on this category since it's also the most generalizable to new modalities. Other methods use output prediction uncertainty for adaptation (Yang et al., 2020; Wang et al., 2021; Roy et al., 2022) or generative training to transform target domain examples or synthesize new ones (Li et al., 2020; Hou & Zheng, 2020; Kurmi et al., 2021; Morerio et al., 2020; Sahoo et al., 2020). Interestingly, Boudiaf et al. (2022) show that previous methods suffer from large hyperparameter sensitivity, and may *degrade* the performance of the source model if not tuned in a scenario-specific manner; an unrealistic setting that violates the assumption of unavailability of labelled target data.

**Test-time Training (TTT).** TTT (Sun et al., 2020) is a related problem where, like in TTA, a pre-trained model is adapted on the target test examples using a self-supervised loss, before making a prediction on those examples. Unlike SFDA and TTA, though, TTT modifies the source training phase by incorporating a similar self-supervised loss there too.

**Domain generalization (DG).** In DG (Wang et al., 2022a), like in SFDA, the target domain is unknown. However, unlike SFDA, no adaptation set is available. Instead the aim is to train a robust source model which works directly on new target distributions. Another important distinction is that DG assumes that information about the source domain is available during deployment on the target domain. A popular strategy for DG is to increase the source model's generalizablity by exposing it to diverse "conditions" at training time via domain randomization (Tobin et al., 2017) or adversarial data augmentation (Volpi et al., 2018; Zhou et al., 2020), or to learn domain-*invariant* representations by training to match all available training "environments" (Arjovsky et al., 2019; Creager et al., 2021), minimizing the worst-case loss over a set of such environments (Sagawa et al., 2020), or decomposing the latent space or model weights into domain-specific and domain-general components (Ilse et al., 2020; Khosla et al., 2012).

## 3 BACKGROUND AND PRELIMINARY INVESTIGATION

### 3.1 PROBLEM FORMULATION

**Notation.** In SFDA for classification, we assume access to a pre-trained model $f_{\boldsymbol{\theta}} : \mathcal{X} \to \mathbb{R}^C$, where $\mathcal{X}$ denotes the input space and $C$ represents the number of classes. This model was trained on a source dataset $\mathbb{D}_s$ sampled from a source distribution $p_s(\mathbf{x})$, and needs to be adapted to a shifted target distribution $p_t(\mathbf{x}) \neq p_s(\mathbf{x})$. We assume we only have access to unlabelled data $\mathbb{D}_t^{\text{adapt}}$ sampled from $p_t$. The goal is to formulate an adaptation procedure $\mathcal{A} : (\boldsymbol{\theta}_s, \mathbb{D}_t^{\text{adapt}}) \to \boldsymbol{\theta}_t$ that produces an adapted version of the original model using the unlabelled dataset. The adapted model's performance is then evaluated on held-out data $\mathbb{D}_t^{\text{test}}$ sampled from $p_t$.

**Single- vs. multi-label classification.** SFDA methods have traditionally addressed single-label classification, in which exactly one category of interest is present in a given sample. Multi-label classification relaxes this assumption by considering that any number of categories (or none) may be present in a given sample, which is common in real-world data. In the single-label case the output probability for sample $\mathbf{x}_i$ is noted $\mathbf{p}_{i,\boldsymbol{\theta}} = \text{softmax}(f_{\boldsymbol{\theta}}(\mathbf{x}_i)) \in [0, 1]^C$. In the multi-label

case, the predictions for each class are treated as separate binary classification problems, resulting in $\mathbf{p}_{i,\boldsymbol{\theta}} = [\sigma(f_{\boldsymbol{\theta}}(\mathbf{x}_i)), 1 - \sigma(f_{\boldsymbol{\theta}}(\mathbf{x}_i))]^\top \in [0,1]^{2 \times C}$, where $\sigma$ is the logistic function.

## 3.2 BIOACOUSTICS TASK

We use *Xeno-Canto* (XC; Vellinga & Planqué, 2015) as the source dataset $\mathbb{D}_s$ for bird species classification in the audio domain. XC is a large collection of user-contributed recordings of wild birds from across the world. Recordings are *focal* (targeted recordings of an individual, captured in natural conditions, as opposed to the *passive* capture of all ambient sounds). Each recording is labeled with the species of the targeted individual; other birds may appear in the recording.

To evaluate adaptation to distribution shifts, we use multiple collections of passive (also called *soundscape*) recordings from various geographical locations as our target datasets. The soundscape datasets exhibit major covariate and label distribution shift from the source dataset. By virtue of being passively recorded, the annotated species are more occluded by environmental noise and distractors. Additionally, the geographical concentration of the datasets means that only a subset of XC's large number of species is present in each dataset, and the label distribution of those species does not necessarily follow that of XC. As a result, models trained on focal recordings have trouble generalizing to soundscapes recordings (Goëau et al., 2018; Kahl et al., 2021).

## 3.3 CATEGORIZATION OF APPROACHES

We will evaluate several methods which have been proposed both for (source-free) domain adaptation on our new shifts. We choose to investigate methods that fit two criteria: generality across tasks and modalities, and high performance. In this section we present our categorization of methods we consider, which we then build upon in Section 4.

**Entropy Minimization (EM).** These methods enforce the *cluster* assumption, i.e., that the boundaries described by the model's head should not cross any high-density region of samples in the feature space. Geometrically, this enforces large margins between the classifier's boundaries and the provided samples by encouraging the model to output increasingly confident predictions for the unlabelled samples (Grandvalet & Bengio, 2004).

We evaluate **TENT** (Wang et al., 2021) as a representative example of the EM approach. TENT adapts the source model by minimizing the entropy of its predictions though tuning the normalization layers' channel-wise scaling and shifting parameters, and updating the layers' population statistics estimates accordingly.

**Teacher-Student (TS).** This is a self-training paradigm where a teacher provides *pseudo-labels* for the unlabelled examples, and a student is then trained to predict them. More formally, TS minimizes

$$\min_{\mathbf{y}_i,...,\mathbf{y}_N,\boldsymbol{\theta}} \ \mathrm{Tr}\left(-\frac{1}{N}\sum_{i=1}^{N} \mathbf{y}_i^\top \log\left(\mathbf{p}_{i,\tau(\boldsymbol{\theta})}\right)\right), \quad \text{s.t} \quad \mathbf{1}^\top \mathbf{y}_i = \mathbf{1}, \ \mathbf{y}_i \geq 0, \tag{1}$$

where $\mathbf{y}_i$ and $\mathbf{p}_{i,\tau(\boldsymbol{\theta})}$ represent the pseudo-label and the model's soft predictions for the sample $\mathbf{x}_i$. The pseudo-label is dependent on the model's weight's, $\boldsymbol{\theta}$, which are transformed using a weight transformation, $\tau$, which is typically set to the identity in TS methods.

Optimization happens in an alternating fashion between student and teacher updates. Specifically, the *teacher-step* minimizes equation 1 w.r.t the pseudo-labels $\{\mathbf{y}_i,...,\mathbf{y}_N\}$ while the *student-step* minimizes equation 1 w.r.t $\boldsymbol{\theta}$ using gradient descent. Different TS methods utilize soft (Xie et al., 2020) or hard (Lee et al., 2013) pseudo-labels.

Intuitively, TS can be seen as an indirect way of minimizing entropy: By training to predict each example's pseudo-label (i.e., its *most likely* label, based on its confidence), the model reinforces its own predictions, thereby increasing its confidence. At the same time, though, it has a *consistency maximization* flavour: Because of the alternation between teacher and student updates, predicting the correct pseudo-label requires consistency throughout time.

We consider two methods from this category in our investigation: First, **pseudo-labelling** (PL; Lee et al., 2013) assigns pseudo-labels to unlabelled examples by picking the maximum-probability

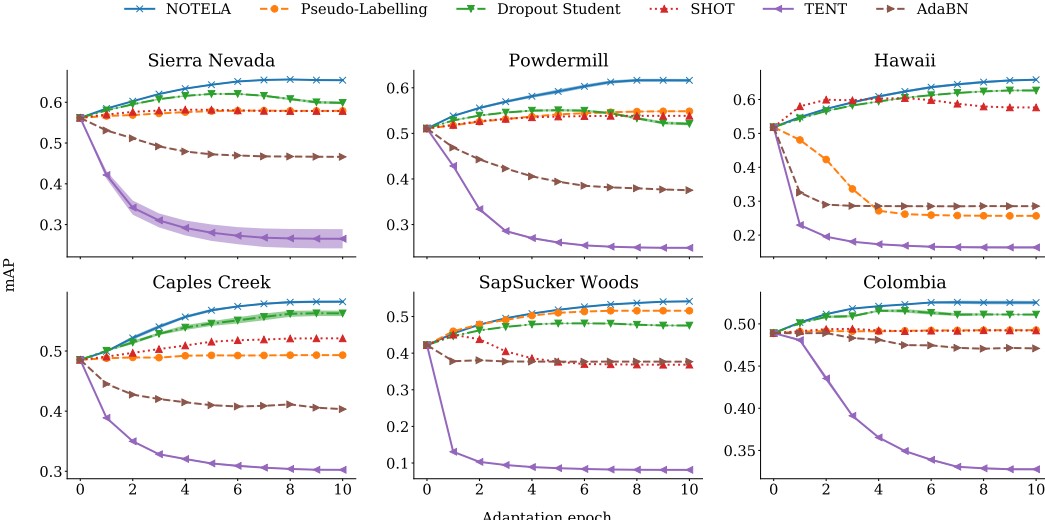

Figure 1: All but one SFDA method fail to consistently improve the source model in terms of $\mathrm{mAP}$ on $\mathbb{D}_t^{test}$ across distribution shifts[1]. Dropout Student succeeds, but only through early stopping, which is infeasible in the SFDA setting. NOTELA achieves consistently stable convergence while improving upon Dropout Student's performance.

class according to the trained model. In the context of SFDA, this translates into assigning pseudo-labels to unlabelled target domain examples using the model trained on the source dataset. Second, **SHOT** (Liang et al., 2020) adapts the feature extractor to the target domain while freezing the classifier head. It performs adaptation through a combination of nearest-centroid pseudo-labelling and an information maximization loss.

**Denoising Teacher-Student (DTS).**   This method builds upon the TS framework by adding some form of "noise" to the student, while keeping the teacher clean. Intuitively, predicting clean pseudo-labels from noisy student predictions leads to maximizing another type of consistency: between different views of the same inputs. Mathematically, DTS differs from TS by setting $\tau \neq \mathrm{Id}$ during the student's forward pass, while keeping it set to the identity during the teacher's forward pass.

Ideas related to DTS have been explored in semi-supervised (Xie et al., 2020; Miyato et al., 2018) and self-supervised learning (Grill et al., 2020; Chen et al., 2020). Notably, Noisy Student (Xie et al., 2020) is a popular representative that we build upon. However, to keep the approach light, both in terms of computation and hyper-parameter load, we consider a simplified model in our investigation, where the same network is used for both the teacher and the student, and dropout is the sole source of noise. We refer to this variant as **dropout student (DS)**. Our proposed model which will be presented later on also falls in this category.

### 3.4 PREVIOUS APPROACHES STRUGGLE ON OUR CHALLENGING SHIFTS

While EM has shown strong performance on various tasks (Wang et al., 2021; Vu et al., 2019), it performed badly on the new shifts we studied. We hypothesize that this is due to a combination of the extremity of distribution shift, as well as the multi-label nature of our classification problem.

Specifically, in the single-label scenario minimizing entropy leads to forcing the model to choose a single class for each example, in an increasingly confident manner. On the other hand, in our multi-label scenario there is no constraint that a class should be chosen. This fact, combined with the potential poor calibration and underconfidence caused by very large distribution shifts (Figure 2, (1,2)) can drive the model to a collapsed state where all class probabilities are zero (Figure 2, (3)).

---

[1]Note that this analysis's purpose is to illustrate the failure modes of SFDA methods; the above plots cannot be used for hyperparameter selection because we are looking at the test sets.

In fact, we find in subsection 5.1 that TENT *underperforms* the baseline of no adaptation on several datasets and hyperparameter settings due to these issues. Later, we will show that both SHOT and pseudo-labelling perform better, but neither systematically prevents the model from degenerating (see Figure 1). In contrast, we find that DTS works significantly better on our challenging shifts, but suffers from stability issues and necessitates a precise early-stopping procedure. This is an important drawback for SFDA, where obtaining a *dataset-specific* (labelled) validation set may not be possible. Motivated by these findings, we next present our approach to SFDA that increases both the performance and stability of DTS methods.

## 4 LAPLACIAN ADJUSTMENT

In this section, we introduce Noisy TEacher student with Laplacian Adjustment (NOTELA), a new method we designed for SFDA, inspired by the desiderata of *stability* and robustness to very large distribution shifts and different problem settings (single- and multi-label, different hyperparameter configurations).

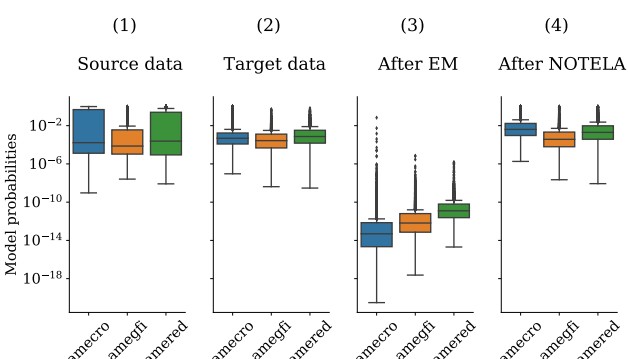

Figure 2: (1) and (2): When shifted to a new domain , the source model exhibits *poorer calibration* than on the original source model, as seen from the lower per-class probability distributions assigned by the model. (3): Applying entropy minimization only worsens the problem, and leads to *general collapse*. (4): Our method largely boosts the source model without affecting model calibration. Species names are represented using their corresponding eBird (Sullivan et al., 2009; 2014) species code.

We begin by exploiting the intuitions obtained from our preliminary investigation. Firstly, EM may not be appropriate in the face of severe distribution shifts which result in very poorly calibrated models. On the other hand, the fact that TS consistently improves upon EM signals that consistency maximization is a useful "auxiliary task", perhaps due to being more robust to miscalibration. Indeed, the fact that DTS further improves, with the additional consistency maximization "task" between clean and noisy views, further strengthens this hypothesis. Taking DTS as our starting place then, we seek to further improve it by utilizing another source of information that we hypothesize is also robust to miscalibration. Specifically, our insight is that, while the head of the model may be poorly calibrated, its *feature space* may still carry useful information about the relationship between examples. In fact, we find that adapting only the linear head using labels on the validation domain described in section 5 boosts the source model by 30% in mAP.

Stemming from this observation, we advocate probing the feature space directly as an *auxiliary source of truth*. Drawing inspiration from classic ideas on manifold regularization (Belkin et al., 2006), NOTELA instantiates this idea by encouraging that nearby points in the feature space be assigned similar pseudo-labels. We hypothesize that this may also help with stability, since the targets that the student is asked to predict will vary less over time, due to the slower-changing similarity in feature space.

**Formalization.** We augment the denoising teacher-student formulation in equation 1 with a Laplacian regularization:

$$\min_{\mathbf{y}_i,\dots,\mathbf{y}_N,\boldsymbol{\theta}} \operatorname{Tr}\left(-\frac{1}{N}\sum_{i=1}^{N}\mathbf{y}_i^\top \log\left(\mathbf{p}_{i,\tau(\boldsymbol{\theta})}\right) + \frac{\alpha}{N}\sum_{i=1}^{N}\mathbf{y}_i^\top \log(\mathbf{y}_i) - \frac{\lambda}{N}\sum_{i=1}^{N}\sum_{j=1}^{N}w_{ij}\,\mathbf{y}_i^\top\mathbf{y}_j\right) \quad (2)$$

There are two changes in Equation 2 compared to Equation 1. First, we introduce a scalar weight $\alpha \in \mathbb{R}$ that controls the softness of pseudo-labels, which we treat as a hyperparameter. Second, we have added the third term that represents a Laplacian regularizer. The value $w_{ij}$ denotes the *affinity* between samples $i$ and $j$ and is obtained by the penultimate layer of the network.

**Optimization.** Disregarding the pairwise Laplacian term allows to directly obtain a closed-form solution to Equation 2, namely $\mathbf{y}_i \propto \mathbf{p}_i^{1/\alpha}$. However, adding the pairwise affinities makes optimization more challenging. We simplify the problem by linearizing the Laplacian term, which allows us to recover a closed-form solution:

$$\mathbf{y}_i \propto \mathbf{p}_i^{1/\alpha} \odot \exp\left(\frac{\lambda}{\alpha}\sum_{j=1}^{N} w_{ij}\mathbf{p}_j\right) . \tag{3}$$

The full proof of Equation 3 can be found in the Appendix. Furthermore, we show in Appendix A.6 that under the assumption of positive semi-definite affinity matrix $(w_{ij})$, Equation 3 becomes an instance of a concave-convex procedure (CCP; Yuille & Rangarajan, 2003).

**Complexity.** Theoretically, the added Laplacian regularization scales quadratically in the number of samples $N$. In practice, we set $w_{ij} = w_{ji} = \frac{1}{d}$ where $0 < d \leq k$ is the number of mutual $k$-nearest neighbours (Brito et al., 1997) of samples $i$ and $j$, and $w_{ij} = w_{ji} = 0$ if these samples are not in each other's $k$-nearest neighbours lists. Finding the $k$-nearest neighbours can be done with $\mathcal{O}(N \log N)$ average time complexity. Equation 3 scales as $\mathcal{O}(NCk)$, but can be fully vectorized across samples.

## 5 EXPERIMENTS

**Data processing and source model.** XC is a large dataset containing a total of 10,932 species. We process XC recordings by resampling the audio to 32 kHz and extracting 6-second slices of relevant audio (see Appendix A.2 for details). We process soundscape recordings by extracting 5-second slices using the provided bounding-box labels (see Appendix A.3 for details).

During training, we take a random 5-second crop and the audio signal's gain is normalized to a random value between 0.15 and 0.25. A technique similar to LEAF (Zeghidour et al., 2021) is used to convert the waveform into a spectrogram. Unlike LEAF, we do not learn a Gaussian lowpass filter and instead apply the Gabor filters with a stride of 320. The resulting output is a power-spectrogram with a time resolution of 100 Hz. For each soundscape target dataset, we use a 75-25% split to obtain $\mathbb{D}_t^{adapt}$, used to adapt the model, and $\mathbb{D}_t^{test}$, used to evaluate the adapted model. These datasets are described in more detail in the appendix.

The most comprehensive bird species classifier we are aware of is BirdNet (Kahl, 2019; Kahl et al., 2021), however the publicly available checkpoints are trained with a combination of focal and soundscape recordings (including many of the test sets considered in this paper),[2] which is incompatible with our SFDA methodology. We instead use an EfficientNet-B1 (Tan & Le, 2019) model we trained ourselves. The output of this model is flattened and projected into a 1280-dimensional embedding space. In addition to species prediction, the model is trained with three auxiliary losses for the bird's order, family, and genus (each having a 0.25 weight).

**Metrics.** We use sample-wise mean average precision (mAP) and class-wise mean average precision (cmAP) for evaluation. Both of these metrics are threshold-free and appropriate for multi-label scenarios. Each can be interpreted as a multi-label generalization of mean reciprocal rank, where ranking of model logits is performed either per-sample (for mAP) or per-class (for cmAP). See subsection A.4 in the appendix for formal definitions.

The mAP metric measures the ability of the model to assign higher logits to any species present in an example. By contrast, cmAP is the mean of the model's per-species classification quality (similar to the average of per-species AUC scores). Note that class-averaging in cmAP corrects for class imbalance, while mAP reflects the natural data distribution. To avoid noisy measurements we only consider species with at least five vocalizations in the dataset when computing cmAP.

**Baselines.** In addition to **TENT**, **pseudo-labelling (PL)**, **SHOT**, and **dropout student (DS)**, which we described in section 3.3, we also consider **AdaBN** (Li et al., 2018). This method recomputes the population statistics of batch normalization (Ioffe & Szegedy, 2015) layers in the pre-trained model using the unlabelled target dataset.

---

[2]Personal correspondence with the authors.

Table 2: Test results on the 6 test target domains (averaged over 5 random seeds) using the optimal hyperparameter configuration found on the validation domain. See Figure 1 for convergence plots.

| Method | S. Nevada | | Powdermill | | Hawai'i | | Caples | | SSW | | Colombia | |
|---|---|---|---|---|---|---|---|---|---|---|---|---|
| | mAP | cmAP | mAP | cmAP | mAP | cmAP | mAP | cmAP | mAP | cmAP | mAP | cmAP |
| Source | 56.2 | 36.4 | 51.1 | 32.2 | 51.8 | 33.2 | 48.5 | 41.2 | 42.2 | 24.7 | 48.9 | 37.7 |
| AdaBN | 46.6 | 34.6 | 37.5 | 29.8 | 28.5 | 27.7 | 40.3 | 38.3 | 37.7 | 25.4 | 47.1 | 36.1 |
| SHOT | 57.8 | 37.8 | 53.8 | 32.5 | 57.6 | 29.0 | 52.1 | 43.0 | 36.8 | 20.0 | 49.2 | 38.1 |
| TENT | 26.5 | 25.2 | 24.9 | 23.9 | 16.4 | 16.1 | 30.2 | 25.4 | 8.1 | 5.8 | 32.8 | 29.3 |
| PL | 57.9 | 38.2 | 54.8 | 32.5 | 25.7 | 22.6 | 49.3 | 42.0 | 51.6 | 29.0 | 49.3 | 38.0 |
| DS | 59.8 | 37.6 | 52.1 | 32.2 | 62.7 | **36.4** | 56.3 | 43.0 | 47.5 | 26.2 | 51.1 | **40.6** |
| NOTELA | **65.4** | **40.0** | **61.6** | **33.7** | **65.8** | 35.4 | **58.2** | **44.2** | **54.1** | **31.2** | **52.5** | **40.6** |

Table 3: Validation and ablation results from the High Sierras bioacoustics dataset.

| Method | mAP | cmAP |
|---|---|---|
| Source | 60.7 | 46.1 |
| AdaBN | 48.4 | 35.8 |
| SHOT | 65.4 | 48.7 |
| TENT | 50.7 | 40.7 |
| PL | 65.2 | 47.9 |
| DS | 72.7 | 47.9 |
| NOTELA | **75.0** | **50.9** |

(a) Best validation results for each method on High Sierras.

| Dropout | Softness | Laplace Reg. | mAP | cmAP |
|---|---|---|---|---|
| | | | 60.7 | 46.0 |
| | ✓ | ✓ | 55.8 | 44.8 |
| ✓ | | ✓ | 70.2 | 48.6 |
| ✓ | ✓ | | 72.7 | 47.9 |
| ✓ | ✓ | ✓ | 75.0 | 50.9 |

(b) Dropout Noise, Softness and Laplacian regularization ingredients **act symbiotically** to provide the best performances. Removing any ingredient leads to significant drops in performances. The first row reflects the source model's performance (no adaptation).

**Hyperparameter selection.** SFDA is fully unsupervised and hence there is no annotated validation set for each target domain. Therefore, we approach hyperparameter selection in a non-standard way: We assume we have one annotated domain which can be used for validation and finding the optimal hyper-parameter configuration. Those hyper-parameters are then fixed, and used to evaluate each method's generalization ability on held-out target domains. We employ the High Sierras dataset for validation and use the remaining domains described in subsection 3.2 for testing only. For every method, we search over hyperparameters such as the learning rate, the subset of parameters to adapt, and whether to use dropout during adaptation. Additionally, we search over specific hyperparameters such as the $\beta$ weight in SHOT, the confidence threshold in PL, or the weights $\{\alpha, \lambda\}$ in NOTELA. The overview of tuned hyperparameters can be found in the appendix.

## 5.1 TAKEAWAYS

**Existing SFDA methods can degrade the baseline.** We found neither AdaBN nor TENT were able to improve the source model, regardless of the hyperparameter configuration or the domain. As for SHOT and PL, we were able to find hyper-parameters that improved over the source model's performance on the validation set, as observed in Table 3a. We observed however that the gains did not consistently translate to the test domains. For example, the PL method was able to boost the source model by $5\%$ mAP on the validation domain, while degrading it by more than $25\%$ mAP on the Hawai'i test domain.

**NOTELA improves stability and performance.** We found that all existing methods displayed a plateau followed by a degradation of the model's performance. This is a serious drawback given the absence of a domain-specific labelled validation set for tuning the training schedule. This is true even for Dropout Student, which was the only previous approach able to consistency outperform the baseline of no adaptation. In contrast, we find that NOTELA not only addresses these stability issues, as demonstrated by Figure 1, but also outperforms all considered baselines, setting the state of the art on our new challenging shifts.

**Laplacian regularization, softness, and noise are symbiotic.** Following our previous observations, we try to gain insights into the relative importance of each ingredient in NOTELA. We find in

Table 4: Top-1 accuracy (averaged over 5 random seeds) on vision test benchmarks. NOTELA approaches best methods on CIFAR-10-C, and surpasses them on ImageNet variants. [†]: Aggregating confidence intervals across the 15 corruptions is not trivial. Instead, we report per-corruption confidence-intervals in Appendix subsection A.7

| Method | CIFAR-10-C[†] (Average across severity 5 corruptions) | ImageNet-R | ImageNet-Sketch |
|---|---|---|---|
| Source | 56.66 | $23.16 \pm 0.0$ | $21.67 \pm 0.0$ |
| AdaBN (Li et al., 2018) | 80.10 | $24.54 \pm 0.08$ | $22.55 \pm 0.14$ |
| SHOT (Liang et al., 2020) | 82.62 | $25.76 \pm 0.15$ | $24.9 \pm 0.75$ |
| TENT (Wang et al., 2021) | 83.41 | $28.55 \pm 0.09$ | $25.04 \pm 0.15$ |
| PL (Lee et al., 2013) | $\mathbf{83.85}$ | $26.16 \pm 1.6$ | $0.76 \pm 0.34$ |
| Dropout Student | 82.17 | $29.15 \pm 0.05$ | $27.68 \pm 0.08$ |
| NOTELA (ours) | 83.67 | $\mathbf{35.23} \pm 0.14$ | $\mathbf{32.50} \pm 0.09$ |

Table 3b that the presence of the noise is crucial, and that removing it leads to worse performances than the non-adapted model. Furthermore, removing softness in labels ($\alpha = 0$) or Laplacian Regularization ($\lambda = 0$) significantly under-performs the full method, thereby highlighting the symbiosis of all three components.

## 5.2 NOTELA PERFORMS STRONGLY ON VISION TASKS

Since our objective is to develop a general approach, we turn back to vision to see how NOTELA fares in that modality.

**Data.** We evaluate on several robustness benchmarks, most of which are used by prior SFDA approaches. First, we use CIFAR-10-C and ImageNet-C (Hendrycks & Dietterich, 2019), a collection of corruptions applied to the CIFAR-10 and ImageNet test sets, spanning 15 corruption types and 5 levels of severity. Second, we use ImageNet-R (Hendrycks et al., 2021), which consists of 30,000 images of 200 of ImageNet's classes obtained by querying for renditions such as "art", "cartoon", "graffiti", etc. Finally, we experiment on *ImageNet-Sketch* (Wang et al., 2019) consisting of 50,000 images constructed by querying Google Images for "sketch of {class}" for all ImageNet classes.

**Models.** We adopt model architectures from previous works, namely a ResNet-50 (He et al., 2016) for ImageNet benchmarks and a Wide ResNet 28-10 (Zagoruyko & Komodakis, 2016) for CIFAR-10 benchmarks. We use the same CIFAR-10 Wide ResNet model checkpoint (provided by Croce et al., 2021) as Wang et al. (2021) for fair comparison. We reimplemented all SFDA methods to ensure they could be tuned and evaluated in the exact same conditions. For each vision dataset we again use a 75-25% split to obtain $\mathbb{D}_t^{adapt}$ and $\mathbb{D}_t^{test}$. We use the most challenging corruption from ImageNet-C (*contrast*) as the validation domain for vision tasks. We report top-1 accuracy for all vision SFDA benchmarks.

**Results.** From Table 4, we find that NOTELA is very competititve on vision benchmarks, despite the change in modality and the single-label setup. It performs on-par with the top performer on CIFAR-10-C, but sets the new state-of-the-art on ImageNet-R and ImageNet-Sketch domains.

## 5.3 CONCLUSION

In this work, we investigate the generalization of recent source-free domain adaptation methods in a new modality to a new and challenging set of distribution shifts: going from focal to passive recordings of wild bird vocalizations for multi-label audio classification. While trying to adapt it, we made the surprising observation that recent approaches sometimes *degrade* the source model. Building on our observations, we proposed an approach called NOTELA, which builds upon Dropout Student by incorporating a Laplacian regularizer when computing the Teacher's pseudo-labels and offering a lightweight optimization procedure. Our proposed approach greatly outperformed existing SFDA approaches on both the studied bioacoustic domains, as well as the established vision SFDA benchmarks. We believe that NOTELA represents a promising first step towards making SFDA approaches applicable to a wide range of modalities and distribution shifts.

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
