# OpenReview forum: "NOTELA: A Generalizable Method for Source Free Domain Adaptation"
_ICLR.cc/2023/Conference — Submitted to ICLR 2023_

### Official Review · Reviewer_mxPp · 2022-10-21

**Confidence:** 4
**Correctness:** 3
**Technical Novelty And Significance:** 2
**Empirical Novelty And Significance:** 1
**Recommendation:** 3

**Clarity, Quality, Novelty And Reproducibility:**

The paper is written well and easy to follow. The quality of the presentation is good but some claims are unfounded (see below). In terms of novelty, the idea is incremental because Laplacian regularization and teach-student networks both have been used for domain adaptation before. In this sense, the paper is not very novel. The code is not included and it is not easy to judge whether the results are reproducible, particularly because the dataset in not common in the area of domain adaptation.

**Strength And Weaknesses:**

Strengths

1. The paper is written well and easy to follow.

2. The paper recognizes a good research question.

Weaknesses:

1. The claims of the paper are unfounded.

2. The idea is incremental.

2. Experiments do not help the evaluation of the proposed method.

**Summary Of The Paper:**

The paper explores the feasibility of source-free domain adaptation in the problem of bird classification using focalized recordings of bird songs as the source domain and the datasets of passive recordings in a target domain. The idea is based on training a teacher-student network paid to predict pseudo-labels for the target domain. Laplacian adjustment then is used to improve the prediction accuracy. Experiments on the bird dataset are provided to demonstrate that the method is effective and outperforms existing methods.

**Summary Of The Review:**

The area of domain adaptation and its instantiations such as source-free formulation is an overcrowded field with many existing works. Hence, simply proposing a new algorithm is not enough. It should be demonstrated that the proposed method is addressing a challenge that existing methods do not address and outperforms existing methods significantly.  Or, a deeper theoretical understanding must be offered. However:

1. The claim that "the standard evaluation protocol only considers a narrow set of shifts in computer vision tasks" is not necessarily true. DomainNet dataset is proposed for exactly the same reason and it has many domains, some with large gaps.

2. The paper has performed experiments on a new dataset. This is great but it makes evaluation difficult. The proposed algorithm should be run on 5-6 comment benchmarks, e.g., Office-Home and DomainNet, that most works in this area use to enable comparison. Currently, the comparison is done with a small number of algorithms. It is not clear how well the proposed method will perform compared to other works not used for comparison. It is also not clear how well the proposed method will work on common benchmarks. For these reasons, I cannot judge how competitive the proposed method is.

3. Experiments should include supervised training on the target domain to enable seeing how challenging the tasks are.

4. The ideas of using Laplacian regularization or teacher-student networks for domain adaptation are not new and have been used in several prior works.

In conclusion, I think this work is not ready for publication and only is a good start.

---

> ### Author Response · Authors · 2022-11-17
> **Authors' reply to reviewer mxPp**
>
> We thank the reviewer for their comments and suggestions. Below, we address the reviewer’s questions and concerns:
>
> *Q: The claim that "the standard evaluation protocol only considers a narrow set of shifts in computer vision tasks" is not necessarily true. DomainNet dataset is proposed for exactly the same reason and it has many domains, some with large gaps.*
>
> Shifts considered to benchmark SFDA methods in computer vision benchmarks are mostly related to style, e.g. sketch/quickdraw/painting/clipart, which we consider relatively narrow. This holds for Visda-C, Office-home and DomainNet.
>
> *Q: The paper has performed experiments on a new dataset. This is great but it makes evaluation difficult. The proposed algorithm should be run on 5-6 comment benchmarks, e.g., Office-Home and DomainNet, that most works in this area use to enable comparison. Currently, the comparison is done with a small number of algorithms. It is not clear how well the proposed method will perform compared to other works not used for comparison. It is also not clear how well the proposed method will work on common benchmarks. For these reasons, I cannot judge how competitive the proposed method is.*
>
> A: Because we use a “clean” validation protocol for all methods, that would require re-running all methods for all those datasets. As mentioned in our previous paragraph, we believe that the shifts those datasets cover are highly correlated. Considering compute constraints, we believe that running methods on a new type of shift, in a new modality, is more informative than running on all existing vision benchmarks. Beyond compute constraints, we also believe that running on a high number of correlated benchmarks may give a false sense of generalizability (for instance, BatchNorm always helps on vision benchmarks, whereas it produces poor results on our new shift).
>
> *Q: Experiments should include supervised training on the target domain to enable seeing how challenging the tasks are.*
>
> A: We thank the reviewer for the suggestion and will include these results in a revised version.
>
> *Q: The ideas of using Laplacian regularization or teacher-student networks for domain adaptation are not new and have been used in several prior works.*
>
> A: We would genuinely appreciate it if the reviewer could provide us with such references.

---

> > ### Comment · Reviewer_mxPp · 2022-11-18
> > **Post-rebuttal Comment**
> >
> > Dear authors,
> >
> > Thank you for your response. After reading your response and the rest of the reviews, I maintain my score because: (i) my concerns are not addressed. For example, some of my concerns could be addressed by providing new experiments but the authors postpone them to the future. (ii) I think some of the concerns raised by the rest of the reviewers are also important and deserve to be addressed. Please also check the following papers:
> >
> > - Donahue, J., Hoffman, J., Rodner, E., Saenko, K. and Darrell, T., 2013. Semi-supervised domain adaptation with instance constraints. In Proceedings of the IEEE conference on computer vision and pattern recognition (pp. 668-675).
> >
> > - Li, J., Seltzer, M.L., Wang, X., Zhao, R. and Gong, Y., 2017. Large-Scale Domain Adaptation via Teacher-Student Learning. Proc. Interspeech 2017, pp.2386-2390.
> >
> > In conclusion, I think the contributions of this work do not rise to an ICLR-level contribution.

---

### Official Review · Reviewer_2cf7 · 2022-10-24

**Confidence:** 3
**Correctness:** 3
**Technical Novelty And Significance:** 2
**Empirical Novelty And Significance:** 2
**Recommendation:** 3

**Clarity, Quality, Novelty And Reproducibility:**

Clarity
The paper was generally clear and easy to follow.

Quality
I think the paper holds good quality. It presents new datasets for evaluating SFDA methods and demonstrates empirical justifications for their motivation and proposed method.

Novelty
The originality of the work is questionable. I think NOTELA holds great similarities with NRC proposed in the previous work (Yang et al., NeurIPS 2021)

Reproducibility
The paper seems to provide enough details to reproduce their experiments. I also expect the authors to open-source their codes in the future for better reproducibility.


**Strength And Weaknesses:**

Strengths
- I like the idea of evaluating the previous SFDA methods on a new set of more challenging naturally-occurring scenarios. I do agree that they should be also working well on other datasets unless they leverage techniques specifically applicable to the computer vision domain.
- I’m not familiar with the audio datasets. However, it seems the proposed audio dataset contains naturally occurring extreme covariates and label shifts. It will provide a good testbed for developing more generalizable SFDA methods.
- I also like how the authors deal with the hyperparameter selection in SFDA. Previous works often neglect the difficulty of the hyperparameter selection and violate the assumption of the unavailability of labeled target data. In contrast, the authors use one dataset for validation and fix the hyperparameters for the other datasets.

Weaknesses
- The authors stated that their insight for the proposed NOTELA is that they can leverage the model’s feature space as another “source of truth”. However, in my view, the motivation does not seem new and it has been the common motivation for numerous previous SFDA methods (Ex. Yang et al., Exploiting the intrinsic neighborhood structure for source-free domain adaptation, NeurIPS 2021.). They assume that the feature space forms meaningful clusters and rely on the clusters to generate pseudo-labels for unlabeled target data. It would be nice if the authors can provide more specific motivations and insights for the proposed model.
- The proposed NOTELA encourages the nearby points in the feature space to be assigned similar pseudo-labels. Specifically, they define an affinity score between samples using the number of mutual k-nearest neighbors. I think NOTELA holds great similarities with NRC proposed in the previous work (Yang et al., Exploiting the intrinsic neighborhood structure for source-free domain adaptation, NeurIPS 2021.). In essence, NRC also defined local affinity with reciprocal nearest neighbors within the feature space and encourage label consistency among data with high local affinity. Please look into the work and explain how the NOTELA is different. Considering the similarities, I think NRC should be also included in all the experiments as a baseline.
- The authors claim that NOTELA is a more generalizable method for SFDA methods. Even though they evaluated the proposed method in some computer vision datasets, they are not widely used datasets for evaluating SFDA methods. To claim that NOTELA is also better in computer vision datasets, I recommend using more widely used vision datasets for evaluating SFDA methods (e.g. OfficeHome and VISDA-C).


**Summary Of The Paper:**

This paper tackles the problem of source-free domain adaptation (SFDA), where a pre-trained source model is adapted using unlabeled target domain data without accessing any source domain data. While previous SFDA algorithms only considered a narrow set of domain shifts in computer vision tasks, the authors put them on a new set of more challenging naturally-occurring scenarios with extreme covariate and label shifts. They propose a new benchmark dataset based on focalized and passive recordings of bird songs. Interestingly, they show that previous SFDA methods underperform in the new dataset. To propose a more generalizable SFDA method, the authors propose NOTELA which enhances the Noisy Student Teacher with a Laplacian Adjustment.

**Summary Of The Review:**

While the paper has its merits, unfortunately, it also has several issues which need to be addressed: (1) unclear motivations for the proposed method, (2) questionable novelty due to the similarities between NOTELA and NRC, and (3) more widely used vision datasets for evaluating SFDA methods.

---

> ### Author Response · Authors · 2022-11-17
> **Authors' reply to reviewer 2cf7**
>
> We thank the reviewer for their comments and suggestions. Below, we address the reviewer’s questions and concerns:
>
> *Q: The proposed NOTELA encourages the nearby points in the feature space to be assigned similar pseudo-labels. Specifically, they define an affinity score between samples using the number of mutual k-nearest neighbors. I think NOTELA holds great similarities with NRC proposed in the previous work (Yang et al., Exploiting the intrinsic neighborhood structure for source-free domain adaptation, NeurIPS 2021.). In essence, NRC also defined local affinity with reciprocal nearest neighbors within the feature space and encourage label consistency among data with high local affinity. Please look into the work and explain how the NOTELA is different. Considering the similarities, I think NRC should be also included in all the experiments as a baseline.*
>
> A: We thank the reviewer for the reference. NOTELA indeed shares a motivation with NRC on using the feature space. Focusing only on the use of nearest-neighbors, NRC presents several heuristic extensions to the vanilla Laplacian regularization (e.g. extended nearest-neighbors), whereas our formulation stays much closer to a purified formulation. Now, from a higher level, our contribution on the method side is not the sole use of Laplacian regularization, but rather the observation that consistency (noisy student) and manifold regularization principles are highly symbiotic, and can be unified in a principled (non-trivial) fashion using our closed-form solution.
>
> *Q: The authors claim that NOTELA is a more generalizable method for SFDA methods. Even though they evaluated the proposed method in some computer vision datasets, they are not widely used datasets for evaluating SFDA methods. To claim that NOTELA is also better in computer vision datasets, I recommend using more widely used vision datasets for evaluating SFDA methods (e.g. OfficeHome and VISDA-C).*
>
> A: Shifts considered to benchmark SFDA methods in computer vision are mostly related to style, e.g. sketch/quickdraw/painting/clipart. Because these shifts are correlated, results also tend to be correlated, such that adding more of such benchmarks only provides limited insights (for instance on our 3 vision benchmarks, the relative ordering of methods remains for the most part similar). In fact, it may even give a false sense of generalizability (for instance, BatchNorm always helps on vision benchmarks, whereas it produces poor results on our new shift).

---

> > ### Comment · Reviewer_2cf7 · 2022-11-18
> > **Post-rebuttal Comment**
> >
> > Thanks for the response, but I'm still leaning toward maintaining my score. While the authors' responses are reasonable in a way, I firmly believe that the authors must provide additional experimental results for a fair comparison with related works. I also agree with other reviewers' comments, which do not seem to be thoroughly addressed in the current manuscript.

---

### Official Review · Reviewer_kHC4 · 2022-10-24

**Confidence:** 5
**Clarity, Quality, Novelty And Reproducibility:** 1. Clarity
**Correctness:** 2
**Technical Novelty And Significance:** 2
**Empirical Novelty And Significance:** 2
**Recommendation:** 3

**Strength And Weaknesses:**

### <Strengths>
1. It is meaningful to study the domain adaptation problem in modalities besides vision and under complex distribution shifts.
2. The self-training framework and the neighborhood consistency term are reasonable and effective.
3. Implementation details are provided for good reproducibility.

### <Weaknesses>
1. The proposed method lacks novelty and seems a combination of techniques in previous works. The dropout-based noisy student is the same as [A] as agreed by the authors. The other part in NOTELA, i.e., the Laplacian regularization term, is very similar to the neighborhood regularization of target data in [B, C].
2. The proposed method requires carefully tuning many hyper-parameters including $\lambda$ and $\alpha$ in Eq. (2). Although the authors use another labeled domain for validation, the condition is not always available and it is common that different target domains prefer different hyper-parameters in realistic scenarios.
3. The problem setting and considered baselines are not clear and even confusing. The authors claim that they consider SFDA under complex distribution shifts with both covariate shifts and label shifts. However, the paper did not consider any popular vision benchmarks to verify the proposed method, including CIFAR10 with curated label shift for domain adaptation under label shift and CIFAR10 with curated label shift, Office-Home, VisDA, DomainNet, and WILDS for SFDA. In addition, the considered baselines are not competitive, even the most up-to-date SHOT and TENT are SFDA works of two years ago. Recent SFDA such as [B, C, D, E] works should be considered for comparison.

References
[A] Self-training with Noisy Student Improves ImageNet Classification, CVPR 2020
[B] Exploiting the Intrinsic Neighborhood Structure for Source-free Domain Adaptation, NeurIPS 2021
[C] Attracting and Dispersing: A Simple Approach for Source-free Domain Adaptation, arxiv 2205.04183
[D] Adaptive Adversarial Network for Source-Free Domain Adaptation, ICCV 2021
[E] Model Adaptation: Historical Contrastive Learning for Unsupervised Domain Adaptation without Source Data, NeurIPS 2021

**Summary Of The Paper:**

This submission studies the source-free domain adaptation (SFDA) problem under both covariate shift and label shift in the audio domain. Specifically, the authors study the task of adapting from focal to passive recordings of wild bird vocalizations for multi-label audio classification. The authors find some SFDA baselines underperform the non-adapted model in the considered task, which motivates them to propose the method NOTELA. NOTELA is based on the noisy student framework and involves two extra terms: encouraging soft prediction and Laplacian regularization. Experiments on the audio task demonstrate NOTELA outperforms other SFDA methods. Experiments on several vision benchmarks also show the efficacy of NOTELA.

**Summary Of The Review:**

This paper targets solving a meaningful and realistic source-free domain adaptation problem for audio tasks with complex distribution shifts. Experiments on the audio benchmarks and vision benchmarks prove the efficacy of the proposed method NOTELA. However, I have two main concerns. The first concern is that NOTELA lacks novelty due to combining existing techniques. The second is that the empirical evaluation is not comprehensive for the problem of source-free domain adaptation.

---

> ### Author Response · Authors · 2022-11-17
> **Authors' reply to reviewer kHC4**
>
> We thank the reviewer for their comments and suggestions. Below, we address the reviewer’s questions and concerns:
>
> *Q: The proposed method lacks novelty and seems a combination of techniques in previous works. The dropout-based noisy student is the same as [A] as agreed by the authors. The other part in NOTELA, i.e., the Laplacian regularization term, is very similar to the neighborhood regularization of target data in [B, C].*
>
> A: Our implementation Dropout Student shares similarities with the Noisy Student, but is not the same, as highlighted in the paper: “we consider a simplified model in our investigation, where the same network is used for both the teacher and the student, and dropout is the sole source of the noise.” In fact, we remove two central pieces of the original method: we do not consider data augmentations for the student (the latter being problem-specific, and the result of years of tuning in vision) and keep the same architecture for the teacher and student.
>
> That being said, we want to emphasize that Laplacian regularization is an old idea to leverage unlabelled data. It was neither introduced by the mentioned paper, nor by us. NOTELA’s novelty lies in (i) the non-trivial observation that both paradigms (Laplacian and consistency) exhibit a strongly symbiotic behavior, and (ii) unifying both paradigms into a single pseudo-labeling scheme, with principled and efficient closed-form solutions.   (iii) observation that this combination leads to a method that generalizes well across problem settings and modalities.
>
> *Q: The proposed method requires carefully tuning many hyper-parameters including lambda and alpha in Eq. (2).*
>
> A: Our method does not require “careful” tuning of hyperparameters. In fact, we found that the fixed set of hyperparameters alpha=1.0 and lamba=1.0 worked best on validation, and worked well across the board.
>
> *Q: Although the authors use another labeled domain for validation, the condition is not always available and it is common that different target domains prefer different hyper-parameters in realistic scenarios.*
>
> A: We clarify that in our experimental protocol, validation hyperparameters are kept frozen across test domains after being found on a held-out domain. Keeping in mind that domain adaptation forbids access to labeled data for a particular target domain of interest, both for training and validation, we emphasize that the ability to work properly across many domains without target-specific hyperparameter tuning is a central challenge for SFDA methods. As agreed by reviewer 2cf7, we believe this challenge was often overlooked in previous worlds, and that our protocol addresses a more realistic setting.
>
> *Q: The problem setting and considered baselines are not clear and even confusing. The authors claim that they consider SFDA under complex distribution shifts with both covariate shifts and label shifts. However, the paper did not consider any popular vision benchmarks to verify the proposed method, including CIFAR10 with curated label shift for domain adaptation under label shift and CIFAR10 with curated label shift, Office-Home, VisDA, DomainNet, and WILDS for SFDA. In addition, the considered baselines are not competitive, even the most up-to-date SHOT and TENT are SFDA works of two years ago. Recent SFDA such as [B, C, D, E] works should be considered for comparison.*
>
> A: We rectify that we employ CIFAR-10C and ImageNet-C which are both used by TENT to benchmark their method, as well as AdaBN and Pseudo-labelling baseline. Note that shifts considered VisDA, DomainNet, Office-Home, etc. are mostly related to style, e.g. sketch/quickdraw/painting/clipart. Because these shifts are correlated, results also tend to be correlated, such that adding more of such benchmarks only provides limited insights (for instance on our 3 vision benchmarks, the relative ordering of methods remains for the most part similar), as compared to comparing on a new shift, in a new modality. In fact, it may even give a false sense of generalizability (for instance, BatchNorm always helps on vision benchmarks, whereas it produces poor results on our new shift).

---

### Official Review · Reviewer_qADF · 2022-10-24

**Confidence:** 2
**Correctness:** 3
**Technical Novelty And Significance:** 4
**Empirical Novelty And Significance:** 3
**Recommendation:** 3

**Clarity, Quality, Novelty And Reproducibility:**

The paper is well written but the motivation is not placed appropriately. The paper seems to indicate that audio datasets would be able to benchmark computer vision methods which is incorrect in my opinion.

The paper contributes a new dataset and a simple method that is able to address the domain gap and perform well.

It should be possible to reproduce the method based on the details from the paper.

**Strength And Weaknesses:**

**Strengths**

* The method identifies potential gaps with existing source free domain adaptation methods.
* The paper proposes a new audio dataset for improving generalizability of existing computer vision methods. The target domains contain significant domain shift in comparison with source domain.
* The proposed method is simple and works on the proposed dataset.

**Weaknesses**

* The paper does not compare with works that have been published in speech domain. For a problem like this, I believe that speech domain is more closely related than vision domain. [ref 1]

* The baselines were developed on the task of computer vision classification. It is not surprising that the methods underperform on this task.

* The creation of domain gap does not seem to be systematic - atleast from the perspective of vision methods. In case of target domain, a sample recording seems to contain multiple birds as opposed to a single bird; while the source domain only has a single bird in the dataset. This does not translate well to methods proposed in vision domain since many of these baseline methods were tested on single class.

* The paper does not seem to compare with more advanced source-free methods that were developed for noise from different classes. There has been some recent work in computer vision literature that tries to induce artificial noise by mixing information from different classes. [ref 2] This would be a better source-free computer vision baseline to compare with since the features are mixed from different classes.


References

[ref 1] Unsupervised Domain Adaptation for Speech Recognition via Uncertainty Driven Self-Training, ICASSP'21

[ref 2] Towards Inheritable Models for Open-Set Domain Adaptation, CVPR'20


**Summary Of The Paper:**

In this paper, the authors try to propose a novel method to the task of source-free domain adaptation. In these settings, the method cannot access the instances from the source domain while adapting to the target domain. The authors find gaps in current computer vision literature by proposing a new dataset in audio domain, where the source domain contains audio recordings with each sample having a recording of a single bird and the target domain contains audio with each sample containing recording of multiple birds in the sample. These recordings are converted into an audio spectogram which is an "image". Therefore, the authors draw comparisons with several computer vision baselines.

**Summary Of The Review:**

This paper makes new contributions to the audio community by proposing a challenging benchmark on bird songs with significant shifts. Further a simple regularizer is proposed which demonstrates improvements over the methods developed for a different computer vision task. Based on the strengths and weaknesses, this paper needs to be revised.

---

> ### Author Response · Authors · 2022-11-17
> **Authors' reply to reviewer qADF**
>
> We thank the reviewer for their comments and suggestions. Below, we address the reviewer’s questions and concerns:
>
> *Q: The paper does not compare with works that have been published in speech domain. For a problem like this, I believe that speech domain is more closely related than vision domain. [ref 1]*
>
> A: We thank the reviewer for the reference. The method developed in the paper only covers the traditional UDA setting, i.e. with the presence of source labels. While an adaptation to the SFDA setting may be possible, it would have to be very opinionated and depart quite significantly from the original method, which we consider beyond the scope of the current paper.
>
> *Q: The baselines were developed on the task of computer vision classification. It is not surprising that the methods underperform on this task.*
>
> A: We kindly disagree with the reviewer. The approaches we benchmark are not computer vision approaches, they are SFDA approaches which were applied to computer vision problems. Nothing about mutual information (SHOT)  or entropy minimization (TENT) is specific to computer vision. We believe it is legitimate to expect that, with adequate hyper-parameter tuning (which we perform), such methods would generalize to the audio domain. Therefore, investigating the success of SFDA approaches on a different modality is a very important contribution to inform the community of the generalizability or robustness issues with our perceived-to-be generic SFDA approaches
>
> *Q:The paper does not seem to compare with more advanced source-free methods that were developed for noise from different classes. There has been some recent work in computer vision literature that tries to induce artificial noise by mixing information from different classes. [ref 2] This would be a better source-free computer vision baseline to compare with since the features are mixed from different classes.*
>
> A: The reference provided addresses the open-set problem, and their feature mixing is only performed to emulate outlier exposure at the training stage, which hopefully leads to better outlier detection once deployed to the “the vendor”. In our setting, we focus on the adaptation stage (as opposed to training) and refrain from making any assumptions on the way the model was trained. The adaptation stage of the mentioned method consists in (weighted) entropy minimization, which in the end appears very similar to the already reproduced TENT baseline.

---

> > ### Comment · Reviewer_qADF · 2022-11-18
> > **Response**
> >
> > I thank the authors for responding to the review. The authors have broadly addressed the concerns presented in the review.  However, I continue to think that this paper would have to establish a better evaluation scheme for fair comparison. I also agree with other reviewers that a stronger benchmarking with other computer vision datasets will help. I am also in agreement with other reviewers that a new dataset is a great contribution; but the evaluation needs to be looked into more carefully to validate insights from the paper.

---

### Decision · Program_Chairs · 2023-01-20

**Decision:**

Reject

**Justification For Why Not Higher Score:**

- Lack of conceptual novelty of the method; it is a straightforward combination of existing techniques
- Lack of comprehensive evaluation on common benchmarks to support the claim that the method is generalizable

**Justification For Why Not Lower Score:**

N/A

**Metareview: Summary, Strengths And Weaknesses:**

This paper studies the source-free domain adaptation problem - it proposes a new algorithm (NOTELA), a new benchmark dataset for evaluation, and a hyperparameter tuning protocol for evaluating such methods.

Reviewers appreciated that the new audio dataset reflects complex real-world domain shifts and that the proposed NOTELA method performs well in the experiments. However, they had major unresolved concerns regarding novelty and comprehensiveness of the empirical evaluations; authors provided a response but did not provide a revision of the paper or additional evaluations. On novelty, reviewers thought that the method is a combination of techniques that have been previously proposed for domain adaptation. On the experiments, the concern was that recent baselines and standard datasets used in the SFDA literature were not included in the comparisons; reviewers thought that claims of generality of the method were not supported due to this lack of evaluation on widely used datasets.

Overall, the paper needs a more comprehensive evaluation to support claims of generality, and a clear discussion of the specific conceptual advances provided by the proposed method before it is ready for publication.